# The aperiodic exponent of subthalamic field potentials reflects excitation/inhibition balance in Parkinsonism

Christoph Wiest[1], Flavie Torrecillos[1], Alek Pogosyan[1], Manuel Bange[2], Muthuraman Muthuraman[2], Sergiu Groppa[2], Natasha Hulse[3], Harutomo Hasegawa[3], Keyoumars Ashkan[3], Fahd Baig[4], Francesca Morgante[4], Erlick A Pereira[4], Nicolas Mallet[5], Peter J Magill[1], Peter Brown[1], Andrew Sharott[1], Huiling Tan[1]*

[1]Medical Research Council Brain Network Dynamics Unit, Nuffield Department of Clinical Neurosciences, University of Oxford, Oxford, United Kingdom; [2]Movement Disorders and Neurostimulation, Biomedical Statistics and Multimodal Signal Processing Unit, Department of Neurology, University Medical Center of the Johannes Gutenberg University Mainz, Mainz, Germany; [3]Department of Neurosurgery, King's College London, London, United Kingdom; [4]Neurosciences Research Centre, Molecular and Clinical Sciences Institute, St. George' s, University of London, London, United Kingdom; [5]Institut des Maladies Neurodégénératives, CNRS UMR5293, Université de Bordeaux, Bordeaux, France

*For correspondence: huiling.tan@ndcn.ox.ac.uk

**Abstract** Periodic features of neural time-series data, such as local field potentials (LFPs), are often quantified using power spectra. While the aperiodic exponent of spectra is typically disregarded, it is nevertheless modulated in a physiologically relevant manner and was recently hypothesised to reflect excitation/inhibition (E/I) balance in neuronal populations. Here, we used a cross-species in vivo electrophysiological approach to test the E/I hypothesis in the context of experimental and idiopathic Parkinsonism. We demonstrate in dopamine-depleted rats that aperiodic exponents and power at 30–100 Hz in subthalamic nucleus (STN) LFPs reflect defined changes in basal ganglia network activity; higher aperiodic exponents tally with lower levels of STN neuron firing and a balance tipped towards inhibition. Using STN-LFPs recorded from awake Parkinson's patients, we show that higher exponents accompany dopaminergic medication and deep brain stimulation (DBS) of STN, consistent with untreated Parkinson's manifesting as reduced inhibition and hyperactivity of STN. These results suggest that the aperiodic exponent of STN-LFPs in Parkinsonism reflects E/I balance and might be a candidate biomarker for adaptive DBS.

## Editor's evaluation

This important work provides compelling evidence for the relationship between aperiodic components of spectral signals in the subthalamic nucleus and changes in neural firing. The manuscript is particularly notable because the authors used a unique cross-species approach in human patients and rats. The mechanistic insight this work provides will be especially impactful given the current interest in considering aperiodic components of electrophysiological signals.

## Introduction

Power spectral densities (PSDs) of neural time series, such as electroencephalography (EEG), electrocorticography (ECoG), or local field potentials (LFPs), tend to follow a 1/f power law distribution, where f is frequency, meaning that power is greatest at low frequencies and diminishes rapidly as frequencies increase (*He et al., 2010*). In this study, we will refer to the slope of the PSD as the aperiodic exponent. For decades, the aperiodic exponent has been deemed unimportant and was often removed from analyses to emphasise brain oscillations (*He, 2014*). Only recently did the aperiodic exponent gain more attention and was attributed a physiological meaning. Aperiodic exponents in LFP, EEG, and ECoG recordings vary by age (*He et al., 2019*; *Schaworonkow and Voytek, 2021*; *Voytek et al., 2015*), cortical depth (*Halgren et al., 2021*), psychiatric disorders (*Adelhöfer et al., 2021*; *Molina et al., 2020*; *Robertson et al., 2019*; *Veerakumar et al., 2019*), and Parkinson's disease (PD) (*Mostile et al., 2019*). They track sensory stimuli (*Waschke et al., 2021*) and change with movement (*Belova et al., 2021*). Moreover, the aperiodic exponent was suggested to reflect neuronal spiking (*Manning et al., 2009*; *Ray and Maunsell, 2011*), synaptic currents (*Baranauskas et al., 2012*; *Buzsáki et al., 2012*), and excitation-inhibition (E/I) balance, which describes the delicate balance of inhibitory and excitatory synaptic inputs to neurons (*Chini et al., 2022*; *Gao et al., 2017*; *Trakoshis et al., 2020*). The latter is supported by multiple studies showing that aperiodic exponents in non-invasive EEG and invasive LFP measurements were modulated by sleep and anaesthesia, with lower exponents, corresponding to slower decay of power with increasing frequencies, during the conscious state, which is linked with increased excitation. Inversely, higher aperiodic exponents, corresponding to faster decay of power with increasing frequencies, were observed during the unconscious state (NREM sleep, anaesthesia), which is linked with more inhibition (*Colombo et al., 2019*; *Huang et al., 2020*; *Lendner et al., 2020*; *Miskovic et al., 2019*; *Niethard et al., 2016*; *Waschke et al., 2021*). In addition, Gao and colleagues found that theta cycle-modulated E/I changes in the rat hippocampus are reflected in the per-cycle PSD exponent of hippocampal LFPs calculated at different theta phases (peaks or troughs) confirming the E/I hypothesis for LFPs (*Gao et al., 2017*).

In our study, we seek to test whether the aperiodic exponent of LFPs recorded in the subthalamic nucleus (STN) tracks changes in E/I balance. To this end, we take advantage of the well-characterised rhythmic changes in neuronal firing patterns within the basal ganglia during slow-wave activity (SWA, see *Figure 1A*) induced in dopamine-depleted rats by general anaesthesia. STN and external globus pallidus (GPe) neurons are rhythmically active in time with cortical SWA in rats rendered Parkinsonian by lesion of midbrain dopamine neurons with 6-hydroxydopamine (6-OHDA) (*Figure 1A*). STN neurons are predominantly active during the active phase of cortical slow (~1 Hz) oscillations (*Magill et al., 2001*; *Mallet et al., 2008a*). Inversely, STN neurons reduce their firing around the inactive phase when GPe-Ti ('type-inactive') neurons tend to fire most. GPe-Ti neurons largely correspond to prototypic GPe neurons that are GABAergic and project to and inhibit STN (*Abdi et al., 2015*; *Mallet et al., 2012*); the GPe is considered the primary source of inhibitory inputs to the STN (*Smith et al., 1998*). Based on single-unit activity, we extracted separate periods with high levels of STN excitation or inhibition, and applied the FoooF algorithm to parameterise PSDs of LFPs during both states (*Donoghue et al., 2020*). Aperiodic exponents of STN-LFPs were consistently modulated with expected differences of the E/I ratio of STN in these states.

In a second step, we tested whether similar findings can be obtained in human STN-LFP recordings from awake patients with PD. In PD, STN shows exaggerated and synchronised oscillatory activity in the beta range (*Bergman et al., 1994*; *Brown et al., 2001*; *Levy et al., 2000*). Dopaminergic medication and deep brain stimulation (DBS) desynchronise STN in the beta range (*Eusebio et al., 2011*; *Kühn et al., 2008*; *Whitmer et al., 2012*) and reduce STN single-unit activity (*Filali et al., 2004*). Here, we compare aperiodic exponents of the STN-LFP ON and OFF dopaminergic medication and ON and OFF DBS. We show that aperiodic exponents of the STN-LFP were increased with both therapies, and these results from PD patients are also consistent with the E/I hypothesis.

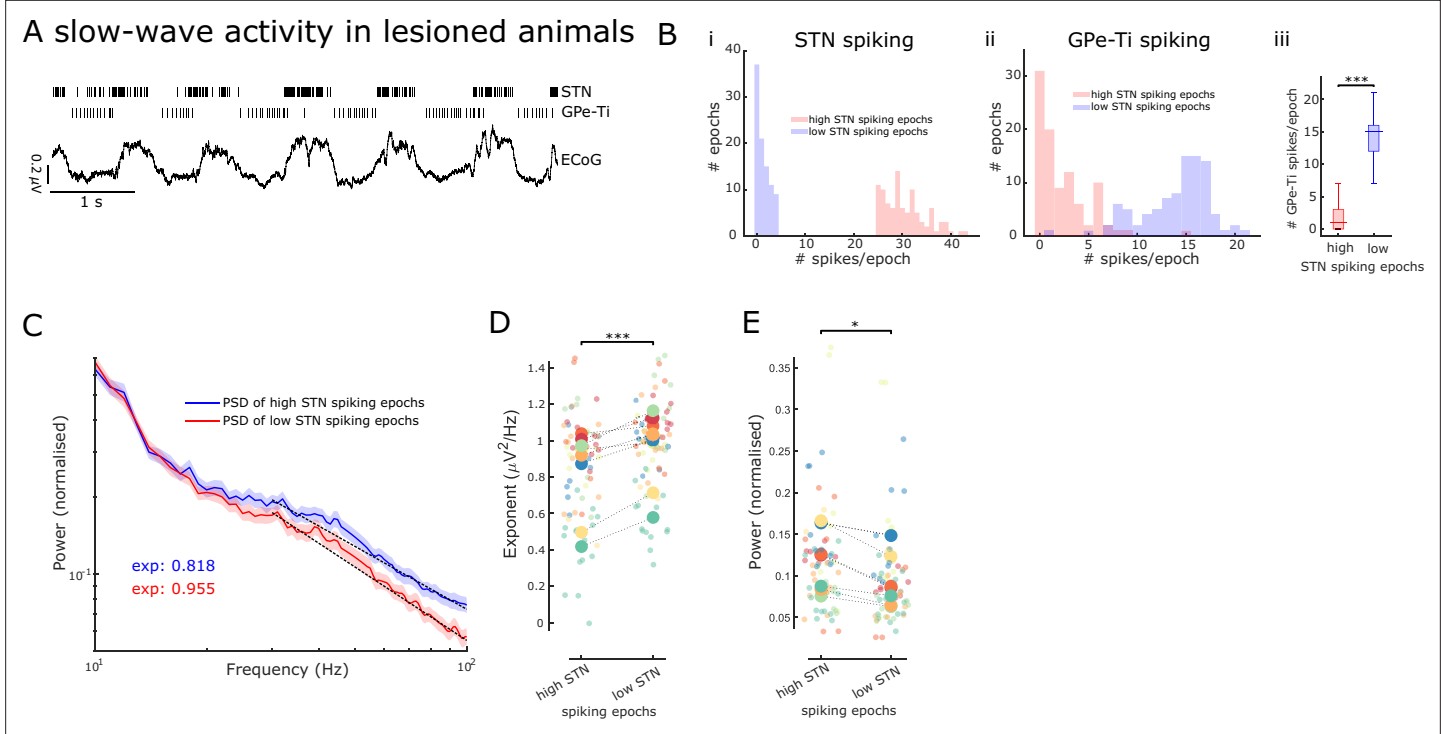

**Figure 1.** Aperiodic exponents and power of subthalamic nucleus-local field potentials (STN-LFPs) between 30 and 100 Hz reflect STN excitation and inhibition in lesioned animals. (**A**) Example traces of electrocorticography (ECoG) and single-unit spiking activity of STN and GPe-Ti neurons during slow-wave activity (SWA) in anaesthetised 6-hydroxydopamine (6-OHDA)-lesioned animals. Note the rhythmic spiking pattern of both neuron types, which either aligns with peaks or troughs of cortical slow (~1 Hz) oscillations. (**B**) We identified 250 ms epochs of relatively high spiking (>75th percentile) and epochs of low spiking (<25th percentile) based on STN neuron activity during cortical SWA. The distribution of STN (i) and GPe-Ti (ii) spiking activity is shown for one example animal, which confirms that spiking rates are clearly differentiable for the two high and low STN spiking states. GPe-Ti neurons are most active when STN neurons are relatively inactive and vice versa (iii; ***p<.001, Wilcoxon rank-sum test). (**C**) Average power spectral densities (PSDs) of STN-LFPs in the two states identified above (mean ± SEM, n = 8 animals). The black dotted lines denote the aperiodic fits (exponent values [exp] are colour-coded) for the respective PSDs between 30 and 100 Hz. (**D**) Aperiodic exponents are high during low STN spiking epochs corresponding to more GPe-Ti activity and, by inference, more inhibition of STN. Inversely, aperiodic exponents are low during high STN spiking epochs associated with less GPe-Ti activity and, by inference, disinhibition (LME: estimate = 0.12, t = 3.92, p<0.001). (**E**) Average power between 30 and 100 Hz in the STN-LFPs is higher when STN neurons are highly active than when STN spiking is low (LME: estimate = –0.02, t = –2.34, p=0.02). (**D, E**) Large dots denote the mean per animal and are colour-coded. Small dots denote individual LFP channels. n = 8 animals, *p<0.05, **p<0.01, ***p<.001, LME.

The online version of this article includes the following figure supplement(s) for figure 1:

**Figure supplement 1.** Signal processing of the electrophysiological data recorded in animals.

**Figure supplement 2.** Knee parameterisation of animal data.

## Results

### Establishing the validity of STN-LFP aperiodic exponents as a marker of E/I balance

To validate the aperiodic exponent of STN activity as an estimate for E/I changes, we first took advantage of the distinct rhythmic changes in neuronal firing patterns within the basal ganglia during SWA in 6-OHDA-lesioned rats under anaesthesia. During SWA, STN neurons exhibit distinct periodic changes in firing rates, which are tightly linked with the phase of cortical slow (~1 Hz) oscillations that dominate SWA (*Figure 1A*; *Magill et al., 2001*). These changes in STN firing rates have been defined, by us and others, in relation to their major excitatory and inhibitory inputs (*Abdi et al., 2015*; *Magill et al., 2001*; *Mallet et al., 2008a*).

To contrast STN activity during opposing conditions of E/I balance, we first distinguished 250 ms epochs of 'high' and 'low' levels of spiking of STN single units during SWA (*Figure 1—figure supplement 1B*). Because STN and GPe-Ti neurons fire in a near anti-phase relationship during cortical SWA (*Mallet et al., 2008a*), high- and low-spiking epochs of STN neurons tend to occur in

time with, respectively, low and high levels of GPe-Ti spiking (*Figure 1B*). We assume that, during low and high levels of GPe-Ti spiking, STN neurons receive low and high levels of inhibitory input from the GPe, respectively. Previous studies have shown that during SWA, excitatory input from cortical projection neurons would be in-phase with the high-spiking epochs of STN neurons, and thus, anti-phase to inhibitory inputs from GPe (*Parr-Brownlie et al., 2007*). First, we computed PSDs of STN-LFPs for the two states (high and low STN spiking epochs, *Figure 1C*) and extracted both aperiodic exponents and average power between 30 and 100 Hz. Aperiodic exponents were smaller in high compared to low STN spiking epochs, consistent with small exponents being generated in conditions of more excitation/less inhibition and vice versa (LME: estimate = 0.12, t = 3.92, p<0.001; *Figure 1D*). The total power of activities between 30 and 100 Hz in STN-LFP was smaller during low STN spiking epochs compared to high STN spiking epochs (LME: estimate = –0.02, t = –2.34, p=0.02; *Figure 1E*). Furthermore, the lack of correlations between power and aperiodic exponents suggests that both power and exponents from the same frequency range are dissociable and might contain different information (Spearman; $\rho$ = –0.07, p=0.56 and $\rho$ = –0.18, p=0.14 for high and low STN spiking epochs, respectively). When pooling aperiodic exponents and power from high and low STN spiking epochs, we still did not observe a significant correlation ($\rho$ = –0.15, p=.07).

Overall, these results from 6-OHDA-lesioned rats suggest that the aperiodic exponent of STN-LFPs is altered according to different levels of STN spiking. The high- and low-spiking epochs of single STN neurons can in turn be ascribed with confidence to an E/I balance that is tipped in favour of excitation and inhibition, respectively. As such, these results support the notion that STN-LFP aperiodic exponents may become valid markers of overall E/I balance in STN.

## Aperiodic exponents from 40 to 90 Hz separate levodopa medication states in PD patients

After corroborating the link between aperiodic exponents in STN-LFPs and changes in E/I balance in the STN in an animal model under anaesthesia, we sought to apply the same concept to clinical data from awake PD patients. According to the classical direct/indirect pathways model of basal ganglia functional organisation, the Parkinsonian STN is overactive (reduced inhibition), which is alleviated by dopaminergic stimulation (*Albin et al., 1989*; *DeLong, 1990*). Therefore, we hypothesised that aperiodic exponents of STN-LFPs are relatively low in the untreated Parkinsonian state and would be increased with dopaminergic medication. To test this hypothesis, we estimated aperiodic exponents in STN-LFPs recorded from 30 hemispheres in 17 patients both ON and OFF levodopa (*Figure 2A and B*). Aperiodic exponents at 40–90 Hz successfully distinguished dopaminergic medication states (paired samples permutation *t*-test; t = –3.31, p=0.0015, Cohen's *d*: 0.60) and were larger ON medication (in 66% of hemispheres), suggesting a faster drop of power with increasing frequencies and in keeping with more inhibition of STN (*Figure 2C*). The levodopa effect on aperiodic exponents was robust against different settings for FoooF parameterisation (*Figure 2—figure supplement 1*). Note that for human data, the frequency range was adjusted to 40–90 Hz (the lower bound due to high-amplitude beta peaks crossing the fitting range and the upper bound due to the harmonic of mains interference; see section 'Signal processing' for more details).

In addition, we quantified periodic beta power identified using the FoooF parameterisation (*Figure 2B*) and average power of six different frequency bands and compared them between conditions ON and OFF levodopa. Periodic beta power differed between the two medication states (paired samples permutation *t*-test, t = 2.08, p=0.048), with higher power in the OFF medication state consistent with previous studies (*Brown et al., 2001*; *Neumann et al., 2017*; *Figure 2C*). Of the six frequency ranges, only beta power distinguished medication states (*Figure 2—figure supplement 2A*). However, neither average (Spearman; $\rho$ = –0.08, p=0.67), low ($\rho$ = –0.20, p=0.28), nor high beta power ($\rho$ = 0.004, p=0.99) correlated with aperiodic exponents (*Figure 2—figure supplement 2B*). In 10 of 20 cases in which the aperiodic exponent was higher ON medication, there was either no clear beta peak in the OFF state or no clear beta reduction with levodopa. This underlines the benefit of multi-biomarker adaptive DBS compared to an algorithm that only relies on beta power as a feedback signal. In 8 of 10 hemispheres in which aperiodic exponents were not higher ON medication, exponents were either very similar and the difference between medication states was <0.1 (four cases) or beta power was not affected by levodopa either ( four cases).

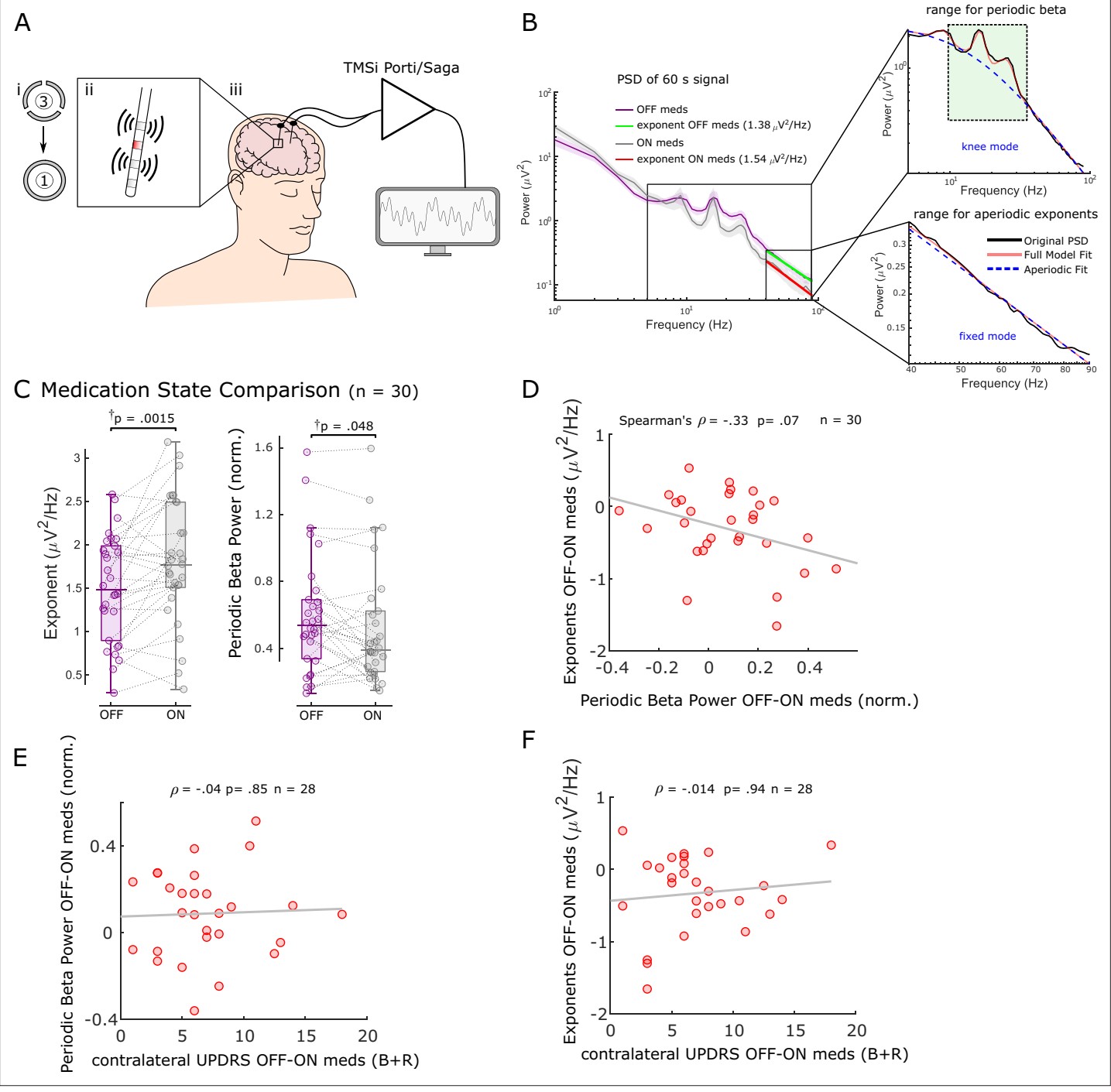

**Figure 2.** Aperiodic exponents between 40 and 90 Hz of subthalamic nucleus-local field potentials (STN-LFPs) distinguish medication states in externalised recordings from Parkinson's patients. (**A**) We analysed 60 s segments of bipolar STN-LFP recorded from Parkinson's patients at rest while leads were externalised (iii). Recordings were performed ON and OFF dopaminergic medication. Directional contacts were fused (i) and bipolar recordings were conducted from contacts adjacent to the stimulation contact (red; ii). (**B**) Average power spectral densities (PSDs) from 30 hemispheres (17 patients) ON and OFF medication of 60 s recordings using Morlet wavelet transforms (mean ± SEM). Aperiodic exponents were computed between 40 and 90 Hz (fixed mode, bottom-right subplot) to avoid high-amplitude beta peaks and harmonics of mains noise at 100 Hz and the fixed mode was used since the PSD was relatively linear in log-log space within this range. To obtain periodic beta power, we fit the FoooF algorithm between 5 and 90 Hz (knee mode, top-right subplot) and picked the power of the largest oscillatory component within the beta range (13–35 Hz, green rectangle). (**C**) Aperiodic exponents and periodic beta power differ between medication states (p=0.0015 and p=0.048). †p-Values were computed using a paired samples permutation *t*-test with multiple comparison correction on 30 hemispheres recorded from 17 Parkinson's disease (PD) patients. (**D**) Aperiodic

*Figure 2 continued on next page*

*Figure 2 continued*

exponent and periodic beta power changes with levodopa are not correlated (Spearman; *ρ* = –0.33, p=0.07, n = 30). (**E, F**) Neither periodic beta power (**E**) nor the aperiodic exponent (**F**) changes with medication are correlated with contralateral appendicular bradykinesia and rigidity UPDRS part III sub-scores OFF-ON levodopa (Spearman: *ρ* = –0.04, p=0.85, n = 28 hemispheres for periodic beta; Spearman: *ρ* = –0.014, p=0.94, n = 28 hemispheres for aperiodic exponents).

The online version of this article includes the following figure supplement(s) for figure 2:

**Figure supplement 1.** Sensitivity analysis of FoooF parameterisation.

**Figure supplement 2.** Spectral changes with levodopa and subthalamic nucleus-deep brain stimulation (STN-DBS).

Medication-induced changes of aperiodic exponents and periodic beta power displayed a trend for a negative correlation (Spearman; *ρ* = –0.33, p=0.07, n = 30; *Figure 2D*), indicating that they may contain similar information on levodopa-induced changes in the STN. This may also hint at a physiological meaning of aperiodic exponents. Furthermore, neither levodopa-induced changes of periodic beta power (Spearman; *ρ* = –0.04, p=0.85, n = 28; *Figure 2E*) nor aperiodic exponents (Spearman; *ρ* = –0.014, p=0.94, n = 28; *Figure 2F*) were correlated with contralateral appendicular bradykinesia and rigidity UPDRS part III sub-scores OFF-ON levodopa in this cohort.

These results suggest that aperiodic exponents of STN-LFPs may be a useful measure to distinguish and track medication states.

## Aperiodic exponents of STN-LFPs from 10 to 50 Hz distinguish periods ON and OFF DBS

High-frequency DBS was thought to induce functional inactivation of the neurons in the stimulated area (*Aziz et al., 1991*; *Benabid et al., 1994*; *Bergman et al., 1990*; *Limousin et al., 1995*). Furthermore, the inactivation hypothesis was corroborated when suppression of neuronal activity was recorded surrounding STN-DBS leads in patients, non-human primates, and rodents (*Filali et al., 2004*; *Meissner et al., 2005*; *Shi et al., 2006*) in line with the classical rate model of movement disorders. Here,

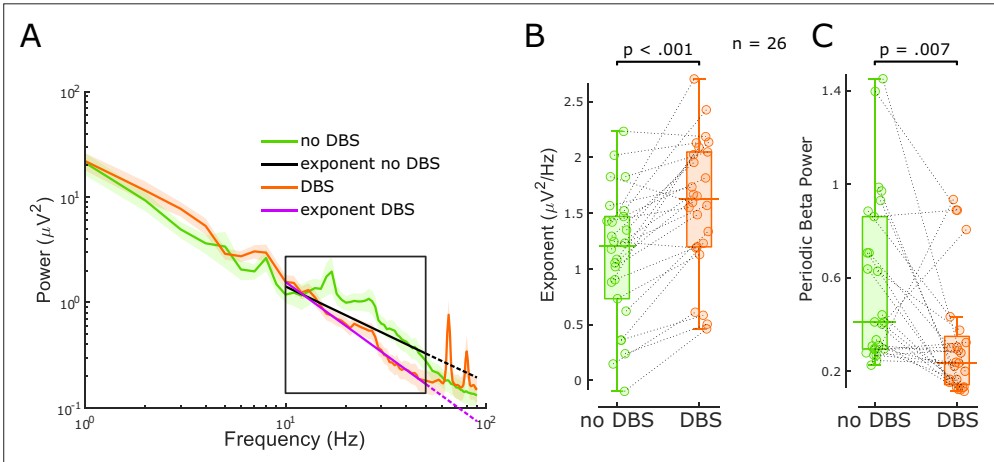

**Figure 3.** The aperiodic exponent between 10 and 50 Hz of subthalamic nucleus-local field potential (STN-LFP) distinguishes stimulation states in externalised recordings from Parkinson's disease (PD) patients. (**A**) Average power spectral densities (PSDs) and linear fitting in the exponential scale of 60 s segments of bipolar STN-LFPs ON and OFF 130 Hz STN-DBS recorded from externalised electrodes in 26 hemispheres from 17 PD patients (mean ± SEM). PSDs during DBS (red) display a spectral plateau >50 Hz, hence, exponents were isolated between 10 and 50 Hz (black rectangle). Linear fitting of the aperiodic exponents between 10 and 50 Hz for no DBS and DBS is shown in black and yellow, respectively. While the dotted extension of the black line is still a relatively good fit for the no DBS PSD, the yellow dotted line deviates from the DBS PSD due to the plateau at around 50 Hz. (**B**) Aperiodic exponents differ between periods of no DBS and periods of 130 Hz STN-DBS (paired samples permutation *t*-test, n = 26, p<0.001). (**C**) Periodic beta power was compared before and during 130 Hz STN-DBS (paired samples permutation *t*-test, n = 26, p=0.007).

The online version of this article includes the following figure supplement(s) for figure 3:

**Figure supplement 1.** Goodness of FoooF parameterisation.

we compared aperiodic exponents at 10–50 Hz of STN-LFPs ON and OFF high-frequency DBS in 26 hemispheres from 17 PD patients. Compared to previous analyses, we lowered the frequency range for parameterisation to avoid a spectral plateau starting >50 Hz when stimulation was on (*Figure 3A*). We found increased aperiodic exponents ON DBS (paired samples permutation *t*-test; t = –6.27, p<0.001, Cohen's *d*: 1.23, *Figure 3B*), a similar effect as levodopa, in keeping with the E/I hypothesis and of the notion that DBS supresses STN firing to some degree.

When the average power of six different frequency ranges and periodic beta power was compared immediately before and during 130 Hz STN-DBS, beta power (primarily high beta) and low gamma power were suppressed with DBS (*Figure 3C*, *Figure 2—figure supplement 2C*) in accordance with our previous work (*Wiest et al., 2020*). The power increase of the high gamma band (51–90 Hz) was driven by artefacts of stimulation in that range (*Figure 3A*). Aperiodic exponent changes with DBS were not correlated with spectral changes of any frequency (*Figure 2—figure supplement 2D*). Again, this implies that the aperiodic exponent is independent of spectral changes averaged over predefined frequency ranges or periodic beta power and might provide additional information.

## Discussion

In this study, we validated the aperiodic exponent of STN-LFPs as a marker of E/I balance using single-unit activity from the basal ganglia in 6-OHDA-lesioned animals, under conditions of where the activity levels of neurons providing major excitatory and inhibitory inputs to STN neurons are well defined. Having validated the approach with single-unit activities under these controlled conditions in rodents, we found that the aperiodic exponent of STN-LFPs recorded from awake PD patients distinguishes medication and stimulation states. Its sensitivity to levodopa and DBS underlines the notion that the aperiodic exponent of the STN-LFP may indicate pathological states of PD and can potentially serve as a feedback signal for adaptive DBS.

### The aperiodic exponent as a marker of E/I balance

Our results support that the aperiodic exponent of STN-LFPs changes with excitatory and inhibitory inputs to STN in both a rodent PD model and patients with PD. STN dendrites are innervated by glutamatergic inputs from ipsilateral somatomotor and frontal cortical areas (*Monakow et al., 1978*), glutamatergic intralaminar thalamic nuclei (*Kita et al., 2016*), and GABAergic inputs from prototypic GPe neurons (*Mallet et al., 2012*). In lesioned animals, on the active phase of the cortical slow oscillation (here referred to as high STN spiking epochs), STN neurons are expected to receive intense excitatory input from cortical and thalamic afferents and decreased inhibitory input from prototypic GPe neurons (*Mallet et al., 2012*), conditions that lead to vigorous STN spiking. In contrast, during the inactive phase of the cortical slow oscillation, inhibition from prototypic GPe neurons should be high and excitation from cortex and thalamus is relatively low, conditions that can completely prevent STN spiking (*Mallet et al., 2012*). We found clear differences in the aperiodic exponents of STN-LFPs between these extreme states of local excitation and inhibition (*Figure 1*).

The majority of previous studies into the relationship between the aperiodic exponent and neuronal activity have been investigated in cortex (*Waschke et al., 2021*). However, the anatomy and physiology of STN differs significantly from cortex in that it is made up of relatively homogenous, glutamatergic projection neurons that fire action potentials autonomously and receive tonic inhibition from prototypic GPe neurons. Cortical circuits, in contrast, comprise a vast array of neuron types and activity of cortical projection neurons is predominantly input driven. Thus, our findings provide important validation that aperiodic exponents reflect changes in E/I balance at the synaptic (LFP) levels, specifically in STN. In addition, in contrast to these previous efforts examining the E/I hypothesis (*Gao et al., 2017*; *Lendner et al., 2020*; *Waschke et al., 2021*), we directly link aperiodic exponents to single-neuron spiking and predefined parameters of the dynamic E/I balance that specifically relate to the majority of neurons in the recorded structure (*Figure 1*). This approach provided a platform for understanding changes in the aperiodic exponent in STN-LFP recordings from humans.

Building on these findings, our clinical results hold up against the E/I hypothesis as well (*Figures 2 and 3*). Aperiodic exponents of STN-LFPs recorded from awake PD patients are increased ON medication and ON DBS when STN local activity is believed to be decreased according to the classical direct/indirect pathways model (*Albin et al., 1989*; *DeLong, 1990*). The increased exponents ON

dopaminergic medication in PD are consistent with a recent study that reported higher aperiodic exponents in dopamine-intact compared to lesioned animals (*Kim et al., 2022*). Furthermore, our results indicate that periodic beta power differs between the two dopaminergic medication states (*Figure 2C*), consistent with previous findings (*Kim et al., 2022*; *Kühn et al., 2006*; *Neumann et al., 2017*; *Ray et al., 2008*). Finally, the missing correlation between medication-induced changes of aperiodic exponents and average beta power suggests that aperiodic exponents capture changes with medication that are not reflected by beta power alone and might be complimentary in an adaptive DBS algorithm ( *Figure 2—figure supplement 2*).

## Can the aperiodic exponent be useful as a feedback signal in adaptive DBS?

Beta-band oscillations were identified as the most promising feedback signal for adaptive DBS considering their strong relationship with motor impairment in PD (Kühn et al., 2006; *Priori et al., 2004*). However, the correlation between beta and contralateral appendicular bradykinesia and rigidity UPDRS scores was not significant in our cohort, and abnormal beta activity is not observed in all patients (*Giannicola et al., 2010*) nor is it correlated with tremor, freezing of gait, or dyskinesia (*Marceglia et al., 2021*). It is therefore unlikely that a beta-only feedback model will address all PD symptoms. Several factors point towards the aperiodic exponent as a candidate feedback signal. First, it changes with medication and DBS (*Figures 2 and 3*), similar characteristics which brought beta oscillations into the spotlight and which may suggest a link to clinical improvement (Kühn et al., 2006; *Kühn et al., 2008*). Second, we observed medication-related changes of the aperiodic exponent in hemispheres without clear beta peaks or medication-related beta changes. Third, we estimate the aperiodic exponent between 40 and 90 Hz, when studying the effect of levodopa, avoiding lower frequencies, which are particularly susceptible to movement-related artefacts (*van Rheede et al., 2022*).

However, there are some obstacles to its use as a feedback marker. First, we could not show a direct link between the aperiodic exponent and clinical symptoms, but a trend for a negative correlation with periodic beta power (*Figure 2D*) and in our data set UPDRS part III scores were not correlated with periodic beta power either (*Figure 2E and F*). To unequivocally assess clinical relevance of aperiodic exponents, motor tests with higher discriminatory power than UPDRS motor sub-scores such as the BRAIN TEST may be useful (*Giovannoni et al., 1999*). Second, the aperiodic exponent would present a feedback signal with low temporal resolution. We averaged PSDs over 60 s to isolate exponents of STN-LFPs. Shortening this time window will introduce noise into the PSD and increase the error of aperiodic estimates. While these temporal dynamics prevent aperiodic exponents from targeting beta bursts, they may provide additional information about the E/I balance with slower temporal dynamics. Third, the frequency range is critical for isolating the aperiodic exponent and results may vary considerably depending on this. Moreover, for some PSDs with large oscillatory components and a spectral plateau at low frequencies, it may even be impossible to obtain an unequivocal linear fit (*Gerster et al., 2022*). Fourth, it is unclear if a miniaturised implantable device will have the computational power to parameterise PSDs in real time and what effects this may have on battery longevity. Whether aperiodic exponents can be combined with other LFP parameters to improve optimal DBS settings will depend on whether they can add additional clinical information than beta power alone, which our results imply (see section 'Aperiodic exponents from 40 to 90 Hz separate levodopa medication states in PD patients'). In this cohort, we did not find correlations between levodopa-induced changes of aperiodic exponents and average beta power of three different frequency ranges (*Figure 2—figure supplement 2B*), but a trend for a negative correlation with periodic beta power, which comprises a measure similar to beta peaks that were used before (*Darcy et al., 2022*; *Neumann et al., 2017*). It is therefore possible that aperiodic exponents extract similar information than beta power, but may aid in subjects where no beta peaks are present or beta peaks are not affected by levodopa.

## Limitations

The chosen frequency band for linear fitting can affect FoooF parameterisation. In the past, many different fitting ranges were applied to extract the aperiodic exponent (reviewed in *Gerster et al., 2022*). Most of these frequency ranges comprise low-frequency oscillations and start in the delta, theta, alpha, or beta range. If prominent low-frequency oscillations are present, this may lead to steepening of the spectrum and error-prone aperiodic fits (*Gerster et al., 2022*). In this study, when

parameterising PSDs from rodent data recorded using high-impedance microelectrodes, we chose the frequency band from 30 to 100 Hz to avoid false-high exponents due to high power in the lower frequency bands. The lower fitting bound of 30 Hz was chosen to exclude overlap of the lower fitting bound with the rising or falling arm of alpha or beta oscillations. A similar frequency range has been recommended to estimate the E/I ratio before (from 30 to 70 Hz, if this range is uncorrupted by oscillatory peaks) (*Gao et al., 2017*; *Trakoshis et al., 2020*). To asses differences with medication in clinical data recorded using macro-electrodes, we narrowed the fitting range to 40–90 Hz to avoid overlap and false-high estimates due to peaks in the high beta range (*Figure 2B*), and to avoid the harmonic of mains interference. These results are robust as long as the selected frequency range covers 40–70 Hz (*Figure 2—figure supplement 1*). For DBS data analysis, we adapted the fitting range to avoid a spectral plateau (*Figure 3A*) which would have resulted in false-low exponents had we used the same fitting range as for the previous analysis. We observed this prominent plateau at frequencies larger than 50 Hz when DBS was switched ON in addition to prominent peaks at half stimulation frequency and other harmonics (*Figure 3A*). However, we did observe reduced power in the high beta and low gamma frequency bands (21–50 Hz) with stimulation, suggesting that the signal-to-noise ratio is sufficient to detect physiological signals in this frequency band. Therefore, we have focused on the lower frequency bands (10–50 Hz) for quantifying aperiodic exponents ON and OFF stimulation. In addition, in our dataset we did not see a link between medication-induced changes of either the aperiodic exponent or periodic beta power and contralateral bradykinesia and rigidity scores OFF-ON levodopa (*Figure 2E and F*) and further studies will be required to investigate if and to what extent aperiodic exponents of STN-LFPs correlate with Parkinsonian symptoms.

## Conclusions

We showed that aperiodic exponents of STN-LFPs reflect STN excitation and inhibition as evinced by single-neuron activity in states of extreme spiking differences in rodents. We further showed that aperiodic exponents of STN-LFP in PD patients are larger ON dopaminergic medication and ON DBS reflecting more inhibition of STN compared to the dopamine-depleted and OFF DBS states. Our results corroborate that the aperiodic exponent contains information with respect to E/I balance and our clinical results give reason to believe the aperiodic component may be useful in a closed-loop feedback algorithm for PD.

## Methods

### Rodent data

Experiments were performed on adult male Sprague–Dawley rats (Charles River) and were conducted in accordance with the Animals (Scientific Procedures) Act, 1986 (UK). Animal data that was analysed in this paper has been generated under the project licence numbers 30/2131 and 30/2629. All details on the 6-OHDA lesion and electrophysiological recordings were published before (*Mallet et al., 2008a*).

### 6-Hydroxydopamine lesion of dopaminergic neurons

Unilateral 6-OHDA lesion was performed as described in *Magill et al., 2001*; *Mallet et al., 2008a*. Then, 25 min before injection, animals received desipramine i.p. to minimise uptake of 6-OHDA by noradrenergic neurons. 6-OHDA was dissolved in NaCl to a concentration of 4 mg/ml of which 3 µl were injected medial to the substantia nigra pars compacta to target the medial forebrain bundle, which results in widespread loss of midbrain dopaminergic neurons. The extent of the dopamine lesion was assessed 14–15 days after injection by apomorphine challenge. Lesions were considered successful if animals made >80 contraversive rotations in 20 min. Electrophysiological recordings were performed ipsilateral to 6-OHDA lesion 21–45 days after surgery when pathophysiological changes in the basal ganglia have likely reached their maxima.

### Electrophysiological recordings

Data was recorded as described in *Mallet et al., 2008a* from eight 6-OHDA-lesioned rats. Here, we summarise key steps of the experimental procedure and data acquisition. Anaesthesia was induced with isoflurane and maintained with urethane, ketamine, and xylazine throughout the recording (*Magill et al., 2006*). Animals were placed in a stereotaxic frame where their body temperature was

maintained at 37°C. ECoG, electrocardiograms (ECG), and respiration rate were monitored to ensure animal well-being. ECoG was recorded via a 1 mm steel screw juxtaposed to the dura mater above the frontal cortex and referenced against another screw in the skull above the ipsilateral cerebellum (*Figure 1—figure supplement 1A*). Raw ECoG traces were bandpass filtered (0.3–1500 Hz) and amplified (2000×) before acquisition. Extracellular recordings of unit activity and LFPs in the external globus pallidus (GPe) and STN were simultaneously made using silicon probes with high-impedance microelectrodes (*Figure 1—figure supplement 1A*). Each probe had one or two vertical arrays (500 µm apart) with 16 recording contacts along the arrays with 100 µm spacing. Monopolar probe signals were referenced against a screw above the contralateral cerebellum. Probes were advanced into the brain under stereotaxic control and extracellular signals were lowpass filtered (6000 Hz). ECoG and probe signals were each sampled at 17.9 kHz using a Power 1401 Analog-Digital converter. Recording locations were verified after experiments using histological procedures. STN was identified by comparison of recorded unit activity with the known characteristics of STN neurons in urethane anaesthesia (*Magill et al., 2001*).

The ECoG measurements were used to assess whether the rodent was in SWA state, which accompanies deep anaesthesia and is similar to activity observed during natural sleep (*Steriade, 2000*). We analysed a total of 28 extracellular LFP channels within STN during SWA (3.5 ± 0.18 channels per animal [mean ± SEM]). STN single-unit activity was isolated from 20 of these channels (2.5 ± 0.07 per animal).

## Signal processing

### Extraction of single-unit activity

Data analysis was performed using custom-written scripts in MATLAB (R2020b). Periods of robust SWA were selected according to previously described characteristics of these brain states (*Magill et al., 2006*; *Magill et al., 2001*). Analyses were performed on 97.61 ± 2.17 s of data during SWA in 6-OHDA-lesioned animals. After offline bandpass filtering (500–6000 Hz) of the probe signals, single-unit activity was isolated with standard spike sorting procedures including template matching, principal component analysis, and supervised clustering (Spike2) as described in *Mallet et al., 2008a*. Isolation of single units was verified by the presence of a distinct refractory period. Only neurons in which <1% of interspike intervals were <2 ms were analysed in this study. For further analysis, single-unit activity was converted such that each spike was represented by a single digital event. To analyse aperiodic exponents of STN-LFPs, raw signals were lowpass filtered at 300 Hz (third-order Butterworth filter; *Oostenveld et al., 2011*). A similar procedure to analyse LFPs from silicon probe electrodes was described before (*Mallet et al., 2008b*).

### Re-referencing

LFPs were downsampled to 2048 Hz. Target channels within STN were re-referenced by subtracting the mean signal across the six neighbouring channels to reduce volume conduction (*Figure 1—figure supplement 1C*). We computed wavelet magnitude squared coherence between the target LFP channel in STN and ECoG to control for volume conduction and opted for the above re-referencing approach with coherence <0.1 for all frequencies >10 Hz (*Figure 1—figure supplement 1C*).

### Spectral decomposition

Spectral parameters were evaluated using continuous complex Morlet wavelet convolution. The entire 100 s time series was decomposed from 1 to 100 Hz with a frequency resolution of 1 Hz (frequencies increased linearly) before epoching to reduce the edge effect on each epoch. Morlet wavelets of 50 cycles were used which did not change as a function of frequency. We counted the spike events per consecutive non-overlapping 250 ms epoch for all unit activity channels (*Figure 1—figure supplement 1B*). The epoch duration was chosen such that it captures the rhythmic dynamics during SWA. For every unit channel, we defined epochs with a spike rate >75th percentile as 'high spiking' and all epochs with a spike rate <25th percentile as 'low spiking' (*Figure 1—figure supplement 1B*). We computed the mean PSD for every 250 ms epoch and normalised it by dividing through the mean power from 1 to 100 Hz (*Figure 1—figure supplement 1D*).

## FoooF parameterisation

For every LFP channel within STN, we computed the mean PSD of all high- and all low-spiking epochs separately. To isolate the aperiodic component from these spectra, we used the open-source FoooF algorithm (version 1.0.0) (*Donoghue et al., 2020*). Settings for the algorithm were set as peak width limits: 2–12; max number of peaks: *infinite*; minimum peak height: 0; peak threshold: 2; and aperiodic mode: *fixed*. Power spectra were parameterised across the frequency range 30–100 Hz. The lower bound was selected to avoid the impact of low-frequency oscillations, the upper bound was selected to avoid the impact of spectral plateaus (*Gerster et al., 2022*). Moreover, PSDs were linear across this frequency range in log-log space assuming a single 1/f like characteristic and did not contain overlapping periodic components (*Figure 1—figure supplement 1E and F*). Hence, higher aperiodic exponents indicate steeper power reduction with increasing frequencies in the PSD and vice versa as shown in *Figure 1—figure supplement 1F*. We evaluated the goodness of fit ($R^2$) for all data used in this study (*Figure 3—figure supplement 1*). In addition, we evaluated and compared the goodness of fit using either '*fixed*' mode or '*knee*' mode in the FoooF fitting. Results show that using the '*knee*' mode for parameterisation did not improve aperiodic fitting (*Figure 1—figure supplement 2*). Therefore, we opted to use the '*fixed*' mode for parameterisation within the selected frequency range. To compare power across different spiking states, we computed the mean power between 30 and 100 Hz of the average PSD of all high or low-spiking epochs.

## Human data

This protocol was approved by the Health Research Authority UK, the National Research Ethics Service local Research Ethics Committee (IRAS: 46576), and the local ethics committee at the University of Mainz (837.208.17 [11042]). Patients were recruited at St. George's University Hospital NHS Foundation Trust, London, King's College Hospital NHS Foundation Trust, London, and the University Medical Center Mainz. Written informed consent, and consent to publish, was obtained before surgery in line with the Declaration of the Principles of Helsinki. We analysed data from 24 patients from these three centres. Data from 17 patients were previously published (*Wiest et al., 2021*; *Wiest et al., 2020*).

### Patients and surgery

Study participants were evaluated by an interdisciplinary team of movement disorder neurologists and functional neurosurgeons and met the UK Parkinson's Disorder Society Brain Bank Diagnostic Criteria for diagnosis of PD. Baseline motor function in the ON and OFF medication state was assessed pre-operatively using the part III of the Unified Parkinson's disease rating scale motor subscale (UPDRS-III). When UPDRS part III scores were reported in *Figure 2*, we used the contralateral appendicular categories (bradykinesia and rigidity scores only). The surgical target was STN. Five models of DBS leads were used: quadripolar (model 3389) or directional (SenSight) leads from Medtronic Inc, Neurological Division, USA, directional (Vercise model DB-2202) or non-directional (model DB-2201) leads from Boston Scientific, USA, and directional leads (Infinity model 6172ANS) from Abbott Inc, USA. DBS implantation was guided either by magnetic resonance imaging alone (St. George's University Hospital) or with additional intra-operative microelectrode recordings and stimulation (King's College Hospital and University Medical Center Mainz). The subthalamic leads were connected to temporary extension leads and these were externalised. Assessment of contact localisation was made through co-registration of immediate post-operative CT with pre-operative MRI by an experienced neurosurgeon or neurologist specialising in deep brain stimulation. Assessment was blinded to the electrophysiological data and made using Lead-DBS (*Horn et al., 2019*).

### Data recording and lead localisation determination

Recordings were made between 3 and 6 days post-operatively, while lead extensions were still externalised and before implantation of the subcutaneous pulse generator. In total, 24 patients (37 hemispheres) were recorded for this study; 17 patients (30 hemispheres) were recorded ON and OFF dopaminergic medication and 17 patients (26 hemispheres) ON and OFF stimulation. In patients with directional leads, the three directional contacts were joined to form a ring contact (*Figure 2Ai*). All LFPs were amplified and sampled at either 2048 Hz using a TMSi Porti (TMS International, Netherlands) or at 4096 Hz using a TMSi Saga32 (TMS International, Netherlands). The ground electrode was placed on the non-dominant forearm. High-frequency stimulation at 130 Hz was only tested at

the middle contacts to allow bipolar LFP recordings from the two surrounding contacts (*Figure 2Aii*). A self-adhesive electrode attached to the patient's back served as a reference for stimulation, which was delivered using a custom-built highly configurable in-house neurostimulator. Stimuli comprised symmetric constant-current biphasic pulses (60 μs pulse width, negative phase first). The stimulation current was started at 0.5 mA and increased in increments of 0.5 mA until first a benefit in Parkinsonian motor symptoms was observed and second side-effect threshold was reached. The contact and current associated with the best clinical improvement were selected. If no stimulation was applied and multiple bipolar configurations were available, we selected the bipolar channel with the largest beta peak at rest. If only one hemisphere was recorded per patient, it was the hemisphere contralateral to the most affected upper limb.

## Signal processing

Analyses were performed on 60 s of data while patients were awake and at rest. LFP time series were highpass filtered at 1 Hz. Complex Morlet wavelet convolution was used for time-frequency decomposition with 50 wavelet cycles between 1 and 90 Hz as described in section 'Spectral decomposition'. To exclude artefacts from mains interference at 50 Hz before isolating the aperiodic component, frequencies between 47 and 53 Hz were removed and the gap was linearly interpolated using the *fillmissing* function. To run the FoooF algorithm, the same settings were used as described in 'FoooF parameterisation', and power spectra were parameterised across the frequency range 40–90 Hz. The lower bound was increased to avoid high-amplitude beta peaks crossing the fitting range (see *Figure 2B*). The upper bound was lowered to avoid the harmonic of mains interference. When a broad frequency band (e.g. 1–100 Hz) was considered, the PSD was not linear in log-log space. However, within the selected frequency band, the power spectrum followed an almost perfect linear line as illustrated in *Figure 2B* (bottom-right plot). Therefore, we used the '*fixed*' mode for parameterisation of the selected frequency range. To isolate periodic beta activity, the PSD was parameterised using FoooF between 5 and 90 Hz ('*knee*' mode in this case as such a wide frequency range is unlikely to only have a single aperiodic component, all other FoooF settings were identical to what was used in animal data). After removing the aperiodic component, the largest peak within the canonical beta range from 8 to 35 Hz was selected (*Figure 2B*). All other FoooF fittings presented elsewhere in this study were performed using the '*fixed*' mode. In the stimulation ON condition, stimulation artefacts led to a plateau at frequencies >50 Hz in the spectra (*Figure 3A*). Therefore, aperiodic exponents were estimated between 10 and 50 Hz to evaluate the effect of DBS. A similar frequency range (5–45 Hz) was used in a previous study (*Chini et al., 2022*). Beta (13–35 Hz), low beta (13–20 Hz), high beta (21–35 Hz), gamma (35–90 Hz), low gamma (35–50 Hz), and high gamma (51–90 Hz) power were computed as the mean power of the normalised (divided by the mean power from 1 to 90 Hz), 50 Hz removed and 1/F-corrected PSD across the respective frequency range.

## Statistics

Statistical analyses were performed using custom-written scripts in MATLAB (R2020b). To perform paired comparisons between the two medication or stimulation conditions, we used a paired samples permutation *t*-test with multiple comparison correction (50,000 permutations each) as implemented in *Groppe, 2022*. When correlations were reported, we calculated Spearman's rank coefficients because the non-baseline-transformed power data are non-normally distributed and contain outliers. When multiple channels within one animal were available, hierarchical comparisons of aperiodic exponents and power across different epochs was performed using linear mixed-effects models to take repeated measures from multiple electrodes for each animal into account. The exponent or power data was set as dependent variable and different STN spiking conditions (e.g. high STN spiking epochs, low STN spiking epochs) as fixed effects. The normal distribution of each variable and the residuals were visually inspected with quantile-quantile plots and histograms of distribution. All models were estimated by the method of maximum likelihood and included random intercept for subjects to allow different intercepts for each subject, thereby capturing individual differences. Multiple statistical tests were performed in this study under FDR control at 5% using the adaptive linear step-up procedure, a modification of the original Benjamini and Hochberg procedure (*Duchet et al., 2021*). This ensures that the expectation of the number of false positives over the total number of positives is less than 5% when many statistical tests are performed. When histograms of distribution are shown, the optimal number

of histogram bins was determined using the Freedman–Diaconis rule (*Freedman and Diaconis, 1981*). All data are shown as mean ± standard error of the mean (SEM) unless mentioned otherwise.

## Sample size estimation, replicates, and group allocation

Due to the explorative nature of this study, we did not perform prospective sample size estimations but included all animal data with wideband recordings from STN that were available to us (n = 8). Since 6-OHDA-lesion was performed unilaterally, only one STN was recorded per animal. Similarly, we did not perform prospective sample size calculations for human data, but included all STN recordings we had available (37 STNs from 24 patients). This approach was chosen as postoperative LFP recordings from DBS electrodes are rare research opportunities. Fortunately, due to the relatively high signal-to-noise ratio of LFP recordings, a small number of patients between 7 and 12 is considered sufficient to detect robust effects in LFP studies (*Alegre et al., 2013*; *Brittain et al., 2012*; *Wessel et al., 2016*).

In our study, each STN was treated as one independent sample. In doing so, we follow the definition that 'biological replicates are parallel measurements of biologically distinct samples that capture random biological variation' (*Blainey et al., 2014*). It should be recognised that the variations we observed in our sample can be caused by variability in the phenotype of PD, of the target location of the electrode, or in the temporary lesion effect of DBS surgery.

This was not a clinical study. We used a within-subject design to test the effects of DBS and dopaminergic medication. All available animal and human data recordings were included in the analysis (no attrition).

## Acknowledgements

This work was supported by the Medical Research Council (MC_UU_00003/2, MC_UU_00003/5, MC_UU_00003/6, MR/V00655X/1, MR/P012272/1), the National Institute for Health Research (NIHR) Oxford Biomedical Research Centre (BRC), Rosetrees Trust, and Parkinson's UK (G-0806).

## Additional information

### Competing interests

Manuel Bange: received grants from the German Research Council (DFG) grant number DFG CRC-TR-128. The author has no other competing interests to declare. Sergiu Groppa: has received funding from BMBF, UM Mainz, Boehringer Foundation, Precisis, DFG, Abbott, Magventure and Innovationsfond GBA. The author received consulting fees from Abbott and Boston Scientific, and received payment or honoraria for lectures from Abbott, Bial, IPSEN, Abbvie, BVDN, BVDN and UCB. The author has no other competing interests to declare. Keyoumars Ashkan: received funding from Medtronic and Abbott, and received support for attending meetings and/or travel from Medtronic and Abbott. The author has no other competing interests to declare. Francesca Morgante: has received research support from NIHR, consulting fees from Boston Scientific and royalties from Springer for the book "Disorders of Movement. The author has received speaking honoraria from Abbvie, Medtronic, Bial, Merz, International Parkinson's disease and Movement Disorder Society, and Advisory board fees from Merz, Abbvie, Boston Scientific. The author has no other competing interests to declare. Erlick A Pereira: has received grants from Life after Paralysis, and royalties and licenses from Elsevier. The author has received consulting fees from Boston Scientific. The author has no other competing interests to declare. Peter J Magill: has received funding from the MRC Programme Grant MC_UU_00003/5 and Parkinson's UK (grant G-0806). The author has no other competing interests to declare. Peter Brown: is a named inventor on the following patent applications: Fischer P He S Tan H Brown P Patent Application: Treatment of gait impairment using deep brain stimulation 2021. WO/2021/250398; Debarros J Brown P Tan H Denison T Patent Application: Emulation of electrophysiological signals derived by stimulation of a body 2020. WO/2020/165591; Debarros J Brown P Tan H Patent Application: Measurement of electrophysiological signals during stimulation of a target area of a body 2020. WO/2020/070492. The author has no other competing interests to declare. Andrew Sharott: has received funding from the MRC Programme Grant MC_UU_00003/6. The author has a pending patent application unrelated to the subject matter of this paper (Patent

WO/2020/183152). The author participates on the Grant Advisory Panel for Aligning Science Across Parkinson's (ASAP). The author has no other competing interests to declare. Huiling Tan: has received funding from the MRC Programme Grant MC_UU_00003/2, the National Institute for Health Research (NIHR) Oxford Biomedical Research Centre (BRC) and Rosetrees Trust. The author is a named inventor in the following patent applications: Fischer P He S Tan H Brown P Patent Application: Treatment of gait impairment using deep brain stimulation 2021. WO/2021/250398; Debarros J Brown P Tan H Denison T Patent Application: Emulation of electrophysiological signals derived by stimulation of a body 2020. WO/2020/165591; Debarros J Brown P Tan H Patent Application: Measurement of electrophysiological signals during stimulation of a target area of a body 2020. WO/2020/070492. The author has no other competing interests to declare. The other authors declare that no competing interests exist.

## Funding

| Funder | Grant reference number | Author |
|---|---|---|
| Medical Research Council | MC_UU_00003/2 | Huiling Tan |
| Medical Research Council | MC_UU_00003/5 | Peter J Magill |
| Medical Research Council | MC_UU_00003/6 | Andrew Sharott |
| National Institute for Health Research Oxford Biomedical Research Centre | | Huiling Tan |
| Rosetrees Trust | | Huiling Tan |
| Parkinson's UK | G-0806 | Peter J Magill |
| Medical Research Council | MR/V00655X/1 | Huiling Tan |
| Medical Research Council | MR/P012272/1 | Huiling Tan |

The funders had no role in study design, data collection and interpretation, or the decision to submit the work for publication.

## Author contributions

Christoph Wiest, Conceptualization, Formal analysis, Investigation, Visualization, Writing - original draft, Writing – review and editing; Flavie Torrecillos, Conceptualization, Supervision, Investigation, Writing – review and editing; Alek Pogosyan, Software; Manuel Bange, Muthuraman Muthuraman, Sergiu Groppa, Data curation; Natasha Hulse, Harutomo Hasegawa, Keyoumars Ashkan, Fahd Baig, Francesca Morgante, Erlick A Pereira, Nicolas Mallet, Resources; Peter J Magill, Resources, Data curation, Funding acquisition, Writing – review and editing; Peter Brown, Conceptualization, Resources, Supervision, Funding acquisition, Project administration; Andrew Sharott, Conceptualization, Resources, Supervision, Funding acquisition, Project administration, Writing – review and editing; Huiling Tan, Conceptualization, Resources, Data curation, Supervision, Funding acquisition, Project administration, Writing – review and editing

## Author ORCIDs

Christoph Wiest http://orcid.org/0000-0001-5728-3951
Muthuraman Muthuraman http://orcid.org/0000-0001-6158-2663
Peter Brown http://orcid.org/0000-0002-5201-3044
Huiling Tan http://orcid.org/0000-0001-8038-3029

## Ethics

This protocol was approved by the Health Research Authority UK, the National Research Ethics Service local Research Ethics Committee (IRAS: 46576) and the local ethics committee at the University of Mainz (837.208.17(11042)). Patients were recruited at St. George's University Hospital NHS Foundation Trust, London, King's College Hospital NHS Foundation Trust, London, and the University Medical Center Mainz. Written informed consent, and consent to publish, was obtained before surgery in line with the Declaration of the Principles of Helsinki. We analysed data from 24 patients from these 3 centres.

Experiments were performed on adult male Sprague Dawley rats (Charles River), and were conducted in accordance with the Animals (Scientific Procedures) Act, 1986 (UK). Animal data that was analysed in this paper has been generated under the project licence numbers 30/2131 and 30/2629. All details on the 6-OHDA lesion and electrophysiological recordings were published before (Mallet et al., 2008a).

## Decision letter and Author response
Decision letter https://doi.org/10.7554/eLife.82467.sa1
Author response https://doi.org/10.7554/eLife.82467.sa2

## Additional files

### Supplementary files
• MDAR checklist

• Source code 1. This source code generates Figures 1-3 of this paper.

### Data availability
The code is included in Source Code 1 and in addition can be found here https://doi.org/10.5287/bodleian:rJ7jyjX97. The animal data used for this project is available at the Medical Research Council Brain Network Dynamics Unit (MRC BNDU) Data Sharing Platform at the University of Oxford https://data.mrc.ox.ac.uk/stn-rat and https://doi.org/10.5287/bodleian:wx6D7oenk. The human data is also available at the MRC BNDU Data Sharing Platform https://data.mrc.ox.ac.uk/stn-lfp-on-off-and-dbs and https://doi.org/10.5287/bodleian:mzJ7YwXvo.

The following datasets were generated:

| Author(s) | Year | Dataset title | Dataset URL | Database and Identifier |
|---|---|---|---|---|
| Wiest C, Torrecillos F, Pogosyan A, Baig F, Pereira E, Morgante F, Ashkan K, Tan H | 2022 | STN local field potential recordings from awake patients with Parkinson's, ON and OFF meds, and during 130 Hz DBS | https://doi.org/10.5287/bodleian:mzJ7YwXvo | Oxford University Research Archive, 10.5287/bodleian:mzJ7YwXvo |
| Wiest C, Tan H | 2022 | Analysis code for publication: 'The aperiodic exponent of subthalamic field potentials reflects excitation/inhibition balance in Parkinsonism' | https://doi.org/10.5287/bodleian:rJ7jyjX97 | Oxford University Research Archive, 10.5287/bodleian:rJ7jyjX97 |
| Mallet N, Sharott A, Brown P, Maghill P | 2022 | Wideband recordings from silicon probes in the subthalamic nucleus of 6-OHDA hemi-lesioned rats during anaesthesia | https://doi.org/10.5287/bodleian:wx6D7oenk | Oxford University Research Archive, 10.5287/bodleian:wx6D7oenk |

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
