## [Editor Report]

This important work provides compelling evidence for the relationship between aperiodic components of spectral signals in the subthalamic nucleus and changes in neural firing. The manuscript is particularly notable because the authors used a unique cross-species approach in human patients and rats. The mechanistic insight this work provides will be especially impactful given the current interest in considering aperiodic components of electrophysiological signals.

---

## [Decision Letter]

**Decision letter after peer review:**

Thank you for submitting your article "The aperiodic exponent of subthalamic field potentials reflects excitation/inhibition balance in Parkinsonism: a cross-species study in vivo" for consideration by *eLife*. Your article has been reviewed by 3 peer reviewers, and the evaluation has been overseen by a Reviewing Editor and Floris de Lange as the Senior Editor. The following individual involved in review of your submission has agreed to reveal their identity: Bradley Voytek (Reviewer #2).

Essential revisions:

1) Our biggest concern was with some of the methodological choices, elaborated on in Reviewer #1's first point. In particular, the rationale with respect to the choice of frequency band used to fit the 1/f, and the inconsistency between experiments was unclear to us. Related, the modeling choices at times fail to account for some aspects of the spectra, which, for some cases, may impact results. The authors should respond to this, either by explaining the rationale or perhaps some re-analysis. One possibility is using a Lorentzian fit ("knee mode"). Please see Reviewer 1's main first point.

2) To elaborate a bit on the point above, as Reviewer #1 highlights in their public review, there was concern that the assumption of a monolithic model of the 1/f pattern (and its representation with a single exponent) fails to capture nuanced/fine-grained aspects of the spectrum, and might confuse an overall elevation of broadband γ power with a change of 1/f exponent. In such cases, observed changes in 1/f exponents would result from improper model fitting and be spurious findings. This issue needs to be carefully addressed.

3) Line 80-83. Is it possible the null result is due to the authors' choice of analyzing the two conditions separately? If the data were pooled between high and low spiking epochs, presumably a significant correlation between power and exponent would be found here. Could the authors please address this?

4) The authors conducted many bivariate correlation analyses, but it would be worthwhile to run a multiple linear regression analysis to see whether and to what degree the different electrophysiological features relate to clinical status.

5) In Figure 3, why do the authors not also show the aperiodic-adjusted β oscillation features?

*Reviewer #1 (Recommendations for the authors):*

Specific issues:

1) Line 5-7, the authors' terminology here confuses the exponent, which is measured from the power spectrum, with the activity that contributes to that power spectrum. In other words, 1/f exponent or aperiodic exponent should not be in the same status as the other terms. It is probably also safe to say that the term "1/f noise" is no longer used in the literature due to the increased recognition that this activity is not noise.

*Reviewer #2 (Recommendations for the authors):*

– The authors perform permutation tests, but report t-values and p-values. Are the p-values the empirical p-values derived from the permutation measure, or are they the inferred p-values from the t-test itself?

– Relatedly, 50,000 permutations seem excessive! Not problematic in any way, just probably overkill.

– The authors should look at Chini et al., *eLife* 2022 as optogenetic support that aperiodic activity indexes EI balance.

– In Figures 1D and E, the individual datapoints are difficult to see. Maybe they can be offset a bit, or the α adjusted so they're not as faint. This is especially true when the figures are printed.

– Across the Figures, the visual language isn't consistent. For example, in Figure 1 red and blue mean low- and high-spiking epochs, respectively. In Figure 2 red and blue mean on and off meds. In Figure 3 red and blue mean on or off DBS. In these cases they're somewhat consistent in the humans (on/off meds is similar to on/off DBS), and theoretically related to low- and high-spiking, but I think different colors should be used to make them better stand apart.

– In Figure 2B, the exponent fit lines in black and yellow are –very– difficult to see, especially in print.

*Reviewer #3 (Recommendations for the authors):*

– The motivation for selecting frequency ranges for analysis is not clear/convincing enough from my point of view, eg. why is it starting at 30 / 40 Hz only?

Potential β band activity below 30 / 40 Hz seems like a weak argument as strong β activity can appear at much lower frequencies or not really at all.

– Chapter 4.2.2: The last sentence appears redundant to me, is it?

– Just a typo on p. 12: SNc needs to be Substantia nigra pars compacta.

– Please double-check the citation Oostenveld et al. 2011. The Journal name is missing.

[Editors' note: further revisions were suggested prior to acceptance, as described below.]

Thank you for resubmitting your work entitled "The aperiodic exponent of subthalamic field potentials reflects excitation/inhibition balance in Parkinsonism" for further consideration by *eLife*. Your revised article has been evaluated by Floris de Lange (Senior Editor) and a Reviewing Editor.

The manuscript has been very much improved but there are some remaining issues that need to be addressed, as outlined below:

Upon discussion between the reviewers we have identified the following points which we would like the authors to address.

1) The reviewers felt the authors had not clearly indicated when a "knee" was included in the fit and when it was not included. The manuscript would be strengthened by clarifying this and also explaining how the decision was made.

2) We are concerned that the authors used the fact that a correlation yielded a p-value greater than 0.05 as justification for the idea that no correlation was present. The authors should include a clear discussion of this and consideration of alternative interpretations.

3) With respect to Figure 2B, we would like the authors to perform an analysis on the narrowband β peak (in addition to the analysis looking at the broader β range already included) and discuss any differences in the on vs off findings.

Below are the individual reviewer comments.

*Reviewer #1 (Recommendations for the authors):*

This revised manuscript is much improved. However, there are still some significant issues that are left unaddressed, which in my opinion precludes publication in the present form.

1. Essential revision Point 1, author point 2. FOOOF fitting with a new parameter.

The authors suggest that "PSDs in log-log space were almost linear (Figure 1C + Figure 2B + Figure 1—figure supplement 1E+F + Figure 3A)." However, a knee is evident in Figure 1C, Figure 2B, and Figure 3A, no-DBS condition. So, this sentence does not fit with the data presented in the figures.

Overall, from reading the response letter, I'm missing a principled approach to decide whether the knee fitting is used or not used in each case. If there was a consistent and objective criterion, the authors should clarify it in the paper.

2. Essential revision Point 3. You can't use a trend-level correlation (p = 0.07) to support a lack of correlation! This correlation is at trend-level, and you might just lack sufficient statistical power for it to cross the arbitrary p = 0.05 threshold.

3. Reviewer #1, point 2. The authors' response here is entirely unconvincing. Their response does not solve the salient issue that Figure 2B and Figure 2C are intuitively contradictory. The authors' response is essentially that a modification of their analysis pipeline yields the same result, even though this result is contrary to the impression one gets from looking at the power spectrum. Instead of believing that the method cannot fail, when the result from a multi-level complicated analysis does not fit the impression from the raw data, it would be prudent to at least consider the possibility that one's tools might be problematic. At the very least, no satisfactory answer is given here about why the statistics do not fit with the raw impression by examining the power spectrum in Figure 2B.

*Reviewer #2 (Recommendations for the authors):*

I've reviewed all of the author's comments and changes, and they have done a fine job addressing all of my concerns.

*Reviewer #3 (Recommendations for the authors):*

In their responses the authors have addressed all my comments convincingly. I also would consider the aspects raised by the other two reviewers as well addressed.

The additional data analyses did not change the nature of the work but confirmed the robustness of the results nicely. The manuscript and the figures have further improved through the revisions. I have no further comments.

---

## [Author Response]

Essential revisions:1) Our biggest concern was with some of the methodological choices, elaborated on in Reviewer #1's first point. In particular, the rationale with respect to the choice of frequency band used to fit the 1/f, and the inconsistency between experiments was unclear to us. Related, the modeling choices at times fail to account for some aspects of the spectra, which, for some cases, may impact results. The authors should respond to this, either by explaining the rationale or perhaps some re-analysis. One possibility is using a Lorentzian fit ("knee mode"). Please see Reviewer 1's main first point.

Choosing an acceptable frequency range is indeed a challenge of quantifying the aperiodic exponent. As the authors of Gerster et al., 2022 put it very accurately: “There is not one-fits-all fitting range”. The choice of the optimal fitting range is mostly driven by prominent oscillatory peaks and spectral plateaus in the PSD. Different electrodes, amplifier and recording systems, as used in this study for animal and human recordings, can also have an impact. Hence, all aperiodic fits were visually inspected as a quality check. In addition, we also tested different fitting ranges to assess how sensitive the results are to changes of the fitting parameters (Figure 2—figure supplement 1).

1. Choice of the fitting range

Despite this heterogeneity in the data, we found fitting ranges in Figure 1 and 2 that largely overlap (30-100 Hz for animal data and 40-90 Hz for the medication comparison in PD patients). We now confirmed that our results in animals data (Figure 1D+E) hold up if the same fitting range is used as for human data (40-90 Hz, shown in Author response image 1). Vice versa, the ON/OFF medication comparison in PD patients holds up if PSDs are parameterised from 30-95 Hz (Figure 2—figure supplement 1A). Harmonics of mains interferences, which were more prominent in human data did not allow to extend the fitting range to 100 Hz. In summary, the results stand for both animal data and human data with a frequency range from 30 to 90 Hz.

**Author response image 1. sa2fig1:** Aperiodic exponent and average γ power changes in animal data between 40 and 90 Hz are consistent with those reported in Figure 1 for a slightly different fitting range.

For the ON/OFF stimulation comparison in human data, we had to adapt the fitting range due to the presence of a spectral flattening/plateau around 50 Hz and prominent stimulation artefacts during stimulation (Figure 3A). Spectral plateaus are regularly present in electrophysiological data and potential sources are discussed in Gerster et al., 2022. Here, we hypothesise that the increased electrical noise floor during stimulation pushed the plateau to frequencies around 50 Hz.

In addition to the spectral plateau around 50 Hz, we could observe artefacts as clear peaks at half stimulation frequency (65 Hz) and some other harmonics in the PSD (e.g. at 80 Hz) at high intensity (Figure 3). While FOOOF will identify these peaks as oscillatory components and exclude them from the aperiodic fit, especially when higher DBS currents are used, there will be many overlapping peaks and little ‘clean’ frequency bands > 50 Hz, which will make a faithful parameterisation impossible (Gerster et al., 2022). Therefore, we would not feel comfortable about any inference from signals above 50 Hz when stimulation is ON.

As we mentioned previously, we are more confident about the signal to noise ratio at frequencies below 50 Hz, since DBS reduces power at 10-50 Hz. Β and low-γ power suppression during DBS are well described in literature (e.g. Wiest et al., 2020). We show in Figure 3A that PSDs of STN-LFPs show a spectral plateau ~ 50 Hz and β/γ power suppression when DBS is switched on. Importantly, power is suppressed in the full 10-50 Hz range and not just a single oscillatory component such as a β peak. The upper frequency bound (at 50 Hz) was determined by the plateau, the lower bound was set at 10 Hz to avoid the effect of low-frequency oscillations (Gerster et al., 2022) after verifying that oscillatory peaks did not overlap with the 10 Hz fitting bound.

2. FOOOF fitting with a knee parameter

We opted against introducing a “knee” in our aperiodic fits, since PSDs in log-log space were almost linear (Figure 1C + Figure 2B + Figure 1—figure supplement 1E+F + Figure 3A). Selecting the “knee mode” will force the algorithm to detect a knee in the given frequency range even when there is none. This will in many cases lead to spurious model fits. To illustrate this, we calculated the PSD of pink noise and performed FOOOF parameterisation with either the settings used in this manuscript (fitting range 40-90 Hz) or knee mode (Author response image 2). While the fixed mode yields an aperiodic exponent close to the ground truth, interpreting the fit results when using knee fits is more complex. When using the knee mode, the exponent reflects the aperiodic component past the knee inflecting point and as shown in this example can then deviate quite considerably from the ground truth (the PSD of pink noise has an exponent of 1). After visual inspection of all PSDs and parameterisations in this study, we decided that the fixed mode will be a good enough fit in the respective fitting ranges and its interpretation as the slope in log-log space is clearer. Also, see a recent study that parameterised power spectra over a frequency range of similar length (4-45 Hz) with the fixed mode (Chini et al., 2022).

**Author response image 2. sa2fig2:** FOOOF parameterisation of pink noise with the same settings used in this manuscript (left) and a knee parameter (right).

2) To elaborate a bit on the point above, as Reviewer #1 highlights in their public review, there was concern that the assumption of a monolithic model of the 1/f pattern (and its representation with a single exponent) fails to capture nuanced/fine-grained aspects of the spectrum, and might confuse an overall elevation of broadband γ power with a change of 1/f exponent. In such cases, observed changes in 1/f exponents would result from improper model fitting and be spurious findings. This issue needs to be carefully addressed.

We agree with the reviewer that broad frequency ranges, such as the one we use to extract periodic β power (5-90 Hz), are unlikely to only have a single aperiodic component. Thus, we now use the knee mode for this frequency range (see Figure 2B).

We changed this in the methods accordingly: “To isolate periodic β activity, the PSD was parameterised using FoooF between 5-90 Hz (knee mode as such a wide frequency range is unlikely to only have a single aperiodic component), otherwise using the same settings, and the largest peak within the canonical β range from 8-35 Hz was selected (Figure 2B).” (lines 419-422)

We have tried to include a knee in the fitting settings for the frequency range from 30 to 100 Hz in Figure 1. However, using the knee mode did not improve the goodness of fit or the fitting error and, in fact, made both parameters slightly worse (Figure 1—figure supplement 2). Based on this, we think the fixed mode would provide the most economical model for the PSDs in Figure 1C. We have now added this analysis in Figure 1—figure supplement 2 to justify the choice of the fixed mode. We added to the manuscript: “Using the ‘*knee*’ mode for parameterisation did not improve aperiodic fitting (Figure 1—figure supplement 2).” (lines 358-359)

As the reviewer pointed out and as we have shown in Figure 1C, broad γ power is increased when STN spiking is higher in animal data, which impacts the aperiodic exponent. Whether this broad γ power increase can be trivially explained by increases in STN spiking is less clear (see comments to reviewer 1). Previous experimental and modelling work showed an overall shift of the spectra to higher power values with increased spiking, while in Figure 1C the spectra are almost identical from 10-20 Hz and only drift apart > 20 Hz. Ray and Maunsell, 2011 show that the positive correlation between LFP power and spiking is less clear for frequencies below 100 Hz. In addition, we do not see any γ power difference between ON and OFF medication states in human data (Figure 2—figure supplement 2A), while aperiodic exponents separate both conditions suggesting that aperiodic exponents reflect more than just γ power changes.

3) Line 80-83. Is it possible the null result is due to the authors' choice of analyzing the two conditions separately? If the data were pooled between high and low spiking epochs, presumably a significant correlation between power and exponent would be found here. Could the authors please address this?

When pooling exponents and power of high and low STN spiking epochs, we still did not find a significant correlation (Spearman; *ρ* = -.15, p = .07 see Author response image 3). We added this information to our manuscript: “When pooling aperiodic exponents and power from high and low STN spiking epochs, we still did not observe a significant correlation (ρ = -.15, p = .07).” (lines 79-81)

This supports that aperiodic exponents and average power of the same frequency range carry complementary information.

**Author response image 3. sa2fig3:** Pooled correlation between power and exponents of high and low STN spiking epochs.

4) The authors conducted many bivariate correlation analyses, but it would be worthwhile to run a multiple linear regression analysis to see whether and to what degree the different electrophysiological features relate to clinical status.

Medication-induced changes of the aperiodic exponent were positively correlated with changes in the γ band (Figure 2—figure supplement 2B), and also showed a tendency for a negative correlation with periodic β power (Figure 2D). However, none of these electrophysiological features related to clinical state. We also used MATLAB’s *fitlm* function to perform a multiple linear regression analysis using the electrophysiological features from Figure 2—figure supplement 2A and Figure 2C (changes between ON and OFF medication) as predictors and the UPDRS score difference between both medication states as response variable (see Author response table 1). Because the R^2^ value of 0.117 is very low, and the p-value of 0.842 is far above the significance level of 0.05, we did not observe a significant linear regression relationship between the response (UPDRS changes with meds) and the predictor variables (electrophysiological differences between medication states). In addition, we tested linear regression models with just a subset of the electrophysiological predictor variables but did not find a significant linear regression relationship (Author response table 1). This is not as surprising as in another manuscript with 106 patients, the highest correlation between medication-induced spectral changes and UPDRS part III scores was in a more finely-tuned low β frequency band (13-20 Hz) with an *ρ* value of 0.361 (Lofredi et al., 2022). The relatively low *ρ* value of this previous study and the lower sample size and the broader β range used in this study may explain the lack of significance in these correlations.

**Author response table 1. sa2table1:** Results of linear regression models.

PredictorVariables	R^2^	p-value
periodic β	0.002	0.821
β (13-35 Hz)	0.027	0.405
γ (15-35 Hz)	0.062	0.203
β + γ + 1/F	0.111	0.411
β + γ	0.071	0.399
all frequencies fromFigure 2—figure supplement 2A + 1/F	0.117	0.842

5) In Figure 3, why do the authors not also show the aperiodic-adjusted β oscillation features?

We have now moved the aperiodic-adjusted β oscillation features with and without DBS from Figure 2—figure supplement 2C to Figure 3C. Periodic β power, like the aperiodic exponent, was modulated by medication and DBS, consistent with previous studies (Brown et al., 2001; Kühn et al., 2008; Whitmer et al., 2012)

Reviewer #1 (Recommendations for the authors):Specific issues:1) Line 5-7, the authors' terminology here confuses the exponent, which is measured from the power spectrum, with the activity that contributes to that power spectrum. In other words, 1/f exponent or aperiodic exponent should not be in the same status as the other terms. It is probably also safe to say that the term "1/f noise" is no longer used in the literature due to the increased recognition that this activity is not noise.

We removed this section of the introduction and replaced it with: “In this study, we will refer to the slope of the PSD as aperiodic exponent.” (lines 4-5)

Reviewer #2 (Recommendations for the authors):– The authors perform permutation tests, but report t-values and p-values. Are the p-values the empirical p-values derived from the permutation measure, or are they the inferred p-values from the t-test itself?

We report the empirical p-values derived from the permutation measure (adjusted for multiple comparisons) as implemented in:

David Groppe (2022). mult_comp_perm_t1(data,n_perm,tail,α_level,mu,reports,seed_state) (https://www.mathworks.com/matlabcentral/fileexchange/29782-mult_comp_perm_t1-data-n_perm-tail-α_level-mu-reports-seed_state), MATLAB Central File Exchange. Retrieved November 8, 2022.

For clarification, we made an addition to the methods section: “To perform paired comparisons between the two medication or stimulation conditions, we used a paired samples permutation t-test with multiple comparison correction (50,000 permutations each) as implemented in (Groppe, 2022).” (lines 431-434)

– Relatedly, 50,000 permutations seem excessive! Not problematic in any way, just probably overkill.

Manly, B.F.J., 1997 suggest using at least 1,000 permutations for a significance level of 0.05 and at least 5,000 permutations for a significance level of 0.01. We chose 50,000 permutations and a significance level of 0.05 as in the example by Groppe 2022. As the reviewer points out, lowering the number of permutations will not affect results.

– The authors should look at Chini et al., eLife 2022 as optogenetic support that aperiodic activity indexes EI balance.

We thank the reviewer for this hint and added Chini et al., 2022 and Trakoshis et al., 2020 in several places throughout the manuscript, for example:

“Moreover, the aperiodic exponent was suggested to reflect neuronal spiking (Manning et al., 2009; Ray and Maunsell, 2011), synaptic currents (Baranauskas et al., 2012; Buzsáki et al., 2012) and excitation-inhibition (E/I) balance, which describes the delicate balance of inhibitory and excitatory synaptic inputs to neurons (Chini et al., 2022; Gao et al., 2017; Trakoshis et al., 2020).” (lines 12-15)

“A similar frequency range (5-45 Hz) was used in a previous study (Chini et al., 2022).” (lines 424-425)

– In Figures 1D and E, the individual datapoints are difficult to see. Maybe they can be offset a bit, or the α adjusted so they're not as faint. This is especially true when the figures are printed.

We have now offset the individual data points and increased the opacity to make them easier to distinguish.

– Across the Figures, the visual language isn't consistent. For example, in Figure 1 red and blue mean low- and high-spiking epochs, respectively. In Figure 2 red and blue mean on and off meds. In Figure 3 red and blue mean on or off DBS. In these cases they're somewhat consistent in the humans (on/off meds is similar to on/off DBS), and theoretically related to low- and high-spiking, but I think different colors should be used to make them better stand apart.

We now changed the colours in Figures 1-3 to clearly distinguish high (blue) and low (red) STN spiking activity (Figure 1), OFF (purple) and ON (grey) levodopa (Figure 2) and OFF (green) and ON (orange) stimulation (Figure 3).

– In Figure 2B, the exponent fit lines in black and yellow are –very– difficult to see, especially in print.

We increased the line width and changed the colours for better contrast.

Reviewer #3 (Recommendations for the authors):– The motivation for selecting frequency ranges for analysis is not clear/convincing enough from my point of view, eg. why is it starting at 30 / 40 Hz only?Potential β band activity below 30 / 40 Hz seems like a weak argument as strong β activity can appear at much lower frequencies or not really at all.

As there is no “one-range-fits-all” frequency range, we adjust the fitting range for parameterisation based on oscillatory components and spectral plateaus of the respective spectra (Gerster et al., 2022). When comparing aperiodic exponents between high and low STN spiking states in rodent data, we chose the 30-100 Hz range. This exact range (Trakoshis et al., 2020) or a similar range (Gao et al., 2017) were used in the past to extract the aperiodic component and to make inferences on the E/I balance. In a complementary analysis, we show in this revision that results are not going to change if the 40-90 Hz range (as applied for clinical data in Figure 2) is chosen for this analysis (Author response image 1). Inversely, our findings in clinical data (Figure 2) hold up if a broader frequency range (30-95 Hz) is selected (Figure 2—figure supplement 1A).

As recommended by Gerster et al., 2022, we chose the lower frequency bound such that it avoids overlap with oscillatory peaks in the α and β range. As can be seen in Author response image 4 for clinical data and Author response image 5 for animal data, such peaks can very well appear between 30 and 40 Hz and therefore affect aperiodic fits. Hence, we decided to avoid overlap with such prominent oscillatory components and increased the lower fitting bound to 30 Hz in animals and 40 Hz in human data. In addition, as suggested by Gerster et al., 2022, we avoided the very low frequency range (1-10 Hz) as this will vary considerably with low frequency oscillations, which dominate the slow-wave sleep in 6-OHDA-lesioned animals (Figure 1A and Magill et al., 2001).

**Author response image 4. sa2fig4:** PSDs from all 30 hemispheres ON and OFF medication. Aperiodic fits are shown between 5-90 Hz (knee mode), which was used to calculate the power of β peaks, and between 40-90 Hz (fixed mode), which was used to estimate the aperiodic exponent of the spectrum.

**Author response image 5. sa2fig5:** Power spectrum of STN LFPs recorded from anaesthetised 6-OHDA-lesioned rats during cortical activation. Note the prominent peak in the β range. The black box designated the frequency range from 30-100 Hz.

The ON and OFF stimulation comparison was performed on the 10-50 Hz frequency range given the prominent spectral plateau that is most likely caused by increased noise in the amplifier/recording system during high-frequency stimulation (Figure 3). In addition, a very similar frequency range (4-45 Hz) was used to extract aperiodic exponents and make inferences on the E/I balance in a recent study Chini et al., 2022.

– Chapter 4.2.2: The last sentence appears redundant to me, is it?

We agree and removed this sentence.

– Please double-check the citation Oostenveld et al. 2011. The Journal name is missing.

Thanks for the hint, we corrected this.

References

Baranauskas, G., Maggiolini, E., Vato, A., Angotzi, G., Bonfanti, A., Zambra, G., Spinelli, A., Fadiga, L., 2012. Origins of 1/f2 scaling in the power spectrum of intracortical local field potential. J. Neurophysiol. 107, 984–994. https://doi.org/10.1152/jn.00470.2011

Brown, P., Oliviero, A., Mazzone, P., Insola, A., Tonali, P., Di Lazzaro, V., 2001. Dopamine dependency of oscillations between subthalamic nucleus and pallidum in Parkinson’s disease. J. Neurosci. 21, 1033–1038. https://doi.org/10.1523/JNEUROSCI.21-03-01033.2001

Buzsáki, G., Anastassiou, C.A., Koch, C., 2012. The origin of extracellular fields and currents — EEG, ECoG, LFP and spikes. Nat. Rev. Neurosci. 13, 407.

Chini, M., Pfeffer, T., Hanganu-Opatz, I., 2022. An increase of inhibition drives the developmental decorrelation of neural activity. *ELife* 11. https://doi.org/10.7554/*eLife*.78811

Donoghue, T., Haller, M., Peterson, E.J., Varma, P., Sebastian, P., Gao, R., Noto, T., Lara, A.H., Wallis, J.D., Knight, R.T., Shestyuk, A., Voytek, B., 2020. Parameterizing neural power spectra into periodic and aperiodic components. Nat. Neurosci. 23, 1655–1665. https://doi.org/10.1038/s41593-020-00744-x

Gao, R., Peterson, E.J., Voytek, B., 2017. Inferring synaptic excitation/inhibition balance from field potentials. Neuroimage 158, 70–78. https://doi.org/10.1016/j.neuroimage.2017.06.078

Gerster, M., Waterstraat, G., Litvak, V., Lehnertz, K., Schnitzler, A., Florin, E., Curio, G., Nikulin, V., 2022. Separating Neural Oscillations from Aperiodic 1/f Activity: Challenges and Recommendations. Neuroinformatics. https://doi.org/10.1007/s12021-022-09581-8

Groppe, D., 2022. mult_comp_perm_t1(data,n_perm,tail,α_level,mu,reports,seed_state) [WWW Document]. MATLAB Cent. File Exch. URL https://www.mathworks.com/matlabcentral/fileexchange/29782-mult_comp_perm_t1-data-n_perm-tail-α_level-mu-reports-seed_state (accessed 11.8.22).

Kim, J., Lee, J., Kim, E., Choi, J.H., Rah, J.-C., Choi, J.-W., 2022. Dopamine depletion can be predicted by the aperiodic component of subthalamic local field potentials. Neurobiol. Dis. 168, 105692. https://doi.org/10.1016/j.nbd.2022.105692

Kühn, A.A., Kempf, F., Brücke, C., Doyle, L.G., Martinez-Torres, I., Pogosyan, A., Trottenberg, T., Kupsch, A., Schneider, G.H., Hariz, M.I., Vandenberghe, W., Nuttin, B., Brown, P., 2008. High-frequency stimulation of the subthalamic nucleus suppresses oscillatory β activity in patients with Parkinson’s disease in parallel with improvement in motor performance. J. Neurosci. 28, 6165–6173. https://doi.org/10.1523/JNEUROSCI.0282-08.2008

Kühn, A.A., Kupsch, A., Schneider, G.H., Brown, P., 2006. Reduction in subthalamic 8-35 Hz oscillatory activity correlates with clinical improvement in Parkinson’s disease. Eur. J. Neurosci. 23, 1956–1960. https://doi.org/10.1111/j.1460-9568.2006.04717.x

Lofredi, R., Okudzhava, L., Irmen, F., Brücke, C., Huebl, J., Krauss, J.K., Schneider, G.-H., Faust, K., Neumann, W.-J., Kühn, A.A., 2022. Subthalamic β bursts correlate with dopamine-dependent motor symptoms in 106 Parkinson’s patients. bioRxiv 2022.05.06.490913. https://doi.org/10.1101/2022.05.06.490913

Manning, J.R., Jacobs, J., Fried, I., Kahana, M.J., 2009. Broadband shifts in local field potential power spectra are correlated with single-neuron spiking in humans. J. Neurosci. 29, 13613–13620. https://doi.org/10.1523/JNEUROSCI.2041-09.2009

Neumann, W.-J., Staub-Bartelt, F., Horn, A., Schanda, J., Schneider, G.-H., Brown, P., Kühn, A.A., 2017. Long term correlation of subthalamic β band activity with motor impairment in patients with Parkinson’s disease. Clin. Neurophysiol. Off. J. Int. Fed. Clin. Neurophysiol. 128, 2286–2291. https://doi.org/10.1016/j.clinph.2017.08.028

Ray, N.J., Jenkinson, N., Wang, S., Holland, P., Brittain, J.S., Joint, C., Stein, J.F., Aziz, T., 2008. Local field potential β activity in the subthalamic nucleus of patients with Parkinson’s disease is associated with improvements in bradykinesia after dopamine and deep brain stimulation. Exp. Neurol. 213, 108–113. https://doi.org/10.1016/j.expneurol.2008.05.008

Ray, S., Maunsell, J.H.R., 2011. Different origins of γ rhythm and high-γ activity in macaque visual cortex. PLoS Biol. 9, e1000610. https://doi.org/10.1371/journal.pbio.1000610

Trakoshis, S., Martínez-Cañada, P., Rocchi, F., Canella, C., You, W., Chakrabarti, B., Ruigrok, A.N., Bullmore, E.T., Suckling, J., Markicevic, M., Zerbi, V., Baron-Cohen, S., Gozzi, A., Lai, M.-C., Panzeri, S., Lombardo, M. V, 2020. Intrinsic excitation-inhibition imbalance affects medial prefrontal cortex differently in autistic men versus women. *ELife* 9. https://doi.org/10.7554/*eLife*.55684

Whitmer, D., de Solages, C., Hill, B., Yu, H., Henderson, J.M., Bronte-Stewart, H., 2012. High frequency deep brain stimulation attenuates subthalamic and cortical rhythms in Parkinson’s disease. Front. Hum. Neurosci. 6, 155. https://doi.org/10.3389/fnhum.2012.00155

Wiest, C., Tinkhauser, G., Pogosyan, A., Bange, M., Muthuraman, M., Groppa, S., Baig, F., Mostofi, A., Pereira, E.A., Tan, H., Brown, P., Torrecillos, F., 2020. Local field potential activity dynamics in response to deep brain stimulation of the subthalamic nucleus in Parkinson’s disease. Neurobiol. Dis. 143, 105019. https://doi.org/10.1016/j.nbd.2020.105019

[Editors' note: further revisions were suggested prior to acceptance, as described below.]

The manuscript has been very much improved but there are some remaining issues that need to be addressed, as outlined below:Upon discussion between the reviewers we have identified the following points which we would like the authors to address.1) The reviewers felt the authors had not clearly indicated when a "knee" was included in the fit and when it was not included. The manuscript would be strengthened by clarifying this and also explaining how the decision was made.

We only used the ‘knee’ mode to isolate periodic β activity in Figure 2B+C. As suggested by Gerster et al., 2022: “If the purpose is not to obtain the 1/f exponent but rather the removal of the aperiodic component for better periodic power assessment, a broadband range (such as 1-100 Hz) should be chosen.” (Gerster et al., 2022) Due to this, we parameterised spectra between 5-90 Hz. This very broad fitting range is unlikely to consist of a single 1/f component (see bend in Figure 2B top right plot and the recommendations for aperiodic component fitting https://fooof-tools.github.io/fooof/auto_tutorials/plot_05-AperiodicFitting.html#sphx-glr-auto-tutorials-plot-05-aperiodicfitting-py) and, hence, we chose the knee parameter.

We clarified this in the revised version of the manuscript: “To isolate periodic β activity, the PSD was parameterised using FoooF between 5-90 Hz (‘knee’ mode in this case as such a wide frequency range is unlikely to only have a single aperiodic component, all other FoooF settings were identical to what was used in animal data). After removing the aperiodic component, the largest peak within the canonical β range from 8-35 Hz was selected (Figure 2B). All other FoooF fittings presented elsewhere in this study were performed using the ‘fixed’ mode.” (lines 436-441)

All other analyses in Figures 1-3 were performed using the ‘fixed’ mode as the PSD within the selected frequency ranges looked linear in log-log space.

In some complementary analyses (in figure supplements) we also tried the knee mode (Figure 2—figure supplement 1 D and Figure 1—figure supplement 2 B) to confirm if the ‘knee mode’ is necessary, and in both cases it is clearly labelled that ‘knee mode’ was used. The results shown in Figure 1—figure supplement 2 suggest that using the ‘knee mode’ did not improve the goodness of fit (quantified by the R^2^ value), and did not reduce the fitting error. These results support the decision of using the ‘fixed’ mode. The rationale for using ‘fixed mode’ can also be found in our manuscript: “Moreover, PSDs were linear across this frequency range in log-log space assuming a single 1/f like characteristic and did not contain overlapping periodic components (Figure 1—figure supplement 1E+F). Hence, higher aperiodic exponents indicate steeper power reduction with increasing frequencies in the PSD and vice versa as shown in Figure 1—figure supplement 1F. We evaluated the goodness of fit (R^2^) for all data used in this study (Figure 3—figure supplement 1). In addition, we evaluated and compared the goodness of fit using either ‘fixed’ mode or ‘knee’ mode in the FoooF fitting. Results show that using the ‘knee’ mode for parameterisation did not improve aperiodic fitting (Figure 1—figure supplement 2). Therefore, we opted to use the ‘fixed’ mode for parameterisation within the selected frequency range.” (lines 365-373)

2) We are concerned that the authors used the fact that a correlation yielded a p-value greater than 0.05 as justification for the idea that no correlation was present. The authors should include a clear discussion of this and consideration of alternative interpretations.

We agree that a p-value of 0.07 shows a trend and cannot be used to justify a lack of correlation. Here, the negative trend is expected given the inverse changes of periodic β power and aperiodic exponents with medication in Figure 2C. We revised the manuscript accordingly: “Medication-induced changes of aperiodic exponents and periodic β power displayed a trend for a negative correlation (Spearman; ρ = -.33, p = .07, n = 30; Figure 2D) indicating that they may contain similar information on levodopa-induced changes in the STN. This may also hint at a physiological meaning of aperiodic exponents.” (lines 122-125)

We deleted the following passage: “… and might add complimentary information to a β activity-based feedback algorithm.” and updated the p-values for correlations between average β power of three different frequency bands and aperiodic exponent changes with levodopa: “However, neither average (Spearman; ρ = -.08, p = .67), low (ρ = -.20, p = .28) nor high β power (ρ = .004, p = .99) correlated with aperiodic exponents (Figure 2—figure supplement 2B).” (lines 112-114)

We also revised our discussion: “First, we could not show a direct link between the aperiodic exponent and clinical symptoms, but a trend for a negative correlation with periodic β power (Figure 2D) and in our data set UPDRS part III scores were not correlated with periodic β power either (Figure 2E+F).” (lines 220-223)

and “In this cohort, we did not find correlations between levodopa-induced changes of aperiodic exponents and average β power of three different frequency ranges (Figure 2—figure supplement 2B), but a trend for a negative correlation with periodic β power, which comprises a measure similar to β peaks that were used before (Darcy et al., 2022; Neumann et al., 2017). It is therefore possible that aperiodic exponents extract similar information than β power, but may aid in subjects where no β peaks are present or β peaks are not affected by levodopa.” (lines 237-242)

3*) With respect to Figure 2B, we would like the authors to perform an analysis on the narrowband β peak (in addition to the analysis looking at the broader β range already included) and discuss any differences in the on vs off findings.*

When analysing periodic β power, we extract a measure that is very similar to narrowband β peaks. As described in the methods, we parameterise the power spectrum between 5 and 90 Hz, subtract the aperiodic component and select the power of the largest β peak between 8 and 35 Hz (largest oscillatory component obtained by FoooF in that range). In Figure 2C we are not showing broad β power, but periodic β power.

This question likely relates to the comment by reviewer 1 that Figures 2B and C seem to contradict each other such that β peaks seem more prominent ON medication in Figure 2B, while their power is higher OFF medication in Figure 2C. The most likely explanation for this disparity is that Figure 2B shows a group average. Plots for individual hemispheres (see Author response image 4) show that the precise frequency of β peaks varies across patients. When averaging across hemispheres, peaks will not line up but merge to a broader β peak (with finer β peaks superimposed, see purple spectrum in Figure 2B). In the ON medication condition, we do not observe the same phenomenon since β peaks are suppressed by levodopa in most hemispheres. However, in some hemispheres β peaks are almost unaffected by medication, which may have caused the prominent β peaks in Figure 2B. When a statistical analysis is performed (Figure 2C), periodic β peak power is larger in the OFF medication state (p = .048). Β peak power is reduced by medication in 18 of 30 hemispheres.

Below are the individual reviewer comments.Reviewer #1 (Recommendations for the authors):This revised manuscript is much improved. However, there are still some significant issues that are left unaddressed, which in my opinion precludes publication in the present form.1. Essential revision Point 1, author point 2. FOOOF fitting with a new parameter.The authors suggest that "PSDs in log-log space were almost linear (Figure 1C + Figure 2B + Figure 1—figure supplement 1E+F + Figure 3A)." However, a knee is evident in Figure 1C, Figure 2B, and Figure 3A, no-DBS condition. So, this sentence does not fit with the data presented in the figures.

We would like to clarify that when we say the ‘PSD in log-log space was almost linear’, we are referring to the PSD in the selected frequency range. We acknowledge that when a very broad frequency range is considered (e.g. 1-100 Hz), there are bends and knees.

We have now clarified this: “When a broad frequency band (e.g. 1-100 Hz) was considered, the PSD was not linear in log-log space. However, within the selected frequency band, the power spectrum followed an almost perfect linear line as illustrated in Figure 2B (bottom right plot). Therefore, we used the ‘fixed’ mode for parameterisation of the selected frequency range.” (lines 433-436)

In Figure 1C, using the ‘knee’ mode actually worsened the goodness of fit and increased the error (Figure 1—figure supplement 2). Furthermore, the potential ‘knee’ in the group plot (Figure 1C) is not apparent when PSDs from all 8 animals are evaluated individually. To illustrate this, we attached see author response image 6. PSDs from 30 to 100 Hz seem almost perfectly linear.

**Author response image 6. sa2fig6:** 

Overall, from reading the response letter, I'm missing a principled approach to decide whether the knee fitting is used or not used in each case. If there was a consistent and objective criterion, the authors should clarify it in the paper.

Please see response to Essential Revisions Point 1.

2. Essential revision Point 3. You can't use a trend-level correlation (p = 0.07) to support a lack of correlation! This correlation is at trend-level, and you might just lack sufficient statistical power for it to cross the arbitrary p = 0.05 threshold.

Please see response to Essential Revisions Point 2.

3. Reviewer #1, point 2. The authors' response here is entirely unconvincing. Their response does not solve the salient issue that Figure 2B and Figure 2C are intuitively contradictory. The authors' response is essentially that a modification of their analysis pipeline yields the same result, even though this result is contrary to the impression one gets from looking at the power spectrum. Instead of believing that the method cannot fail, when the result from a multi-level complicated analysis does not fit the impression from the raw data, it would be prudent to at least consider the possibility that one's tools might be problematic. At the very least, no satisfactory answer is given here about why the statistics do not fit with the raw impression by examining the power spectrum in Figure 2B.

Please see response to Essential Revisions Point 3. We do not argue that a modification of our analysis pipeline yields the same result, but that the visual illusion that is described by the reviewer is due to the summing of β peaks at different frequencies in the OFF medication state.

References

Darcy, N., Lofredi, R., Al-Fatly, B., Neumann, W.-J., Hübl, J., Brücke, C., Krause, P., Schneider, G.-H., Kühn, A., 2022. Spectral and spatial distribution of subthalamic β peak activity in Parkinson’s disease patients. Exp. Neurol. 114150. https://doi.org/10.1016/j.expneurol.2022.114150

Gerster, M., Waterstraat, G., Litvak, V., Lehnertz, K., Schnitzler, A., Florin, E., Curio, G., Nikulin, V., 2022. Separating Neural Oscillations from Aperiodic 1/f Activity: Challenges and Recommendations. Neuroinformatics. https://doi.org/10.1007/s12021-022-09581-8

Neumann, W.-J., Staub-Bartelt, F., Horn, A., Schanda, J., Schneider, G.-H., Brown, P., Kühn, A.A., 2017. Long term correlation of subthalamic β band activity with motor impairment in patients with Parkinson’s disease. Clin. Neurophysiol. Off. J. Int. Fed. Clin. Neurophysiol. 128, 2286–2291. https://doi.org/10.1016/j.clinph.2017.08.028